# Mott gap collapse in lightly hole-doped Sr$_{2-x}$K$_x$IrO$_4$

J. N. Nelson [1], C. T. Parzyck [1], B. D. Faeth[1], J. K. Kawasaki [1,2,3,4], D. G. Schlom [2,3] & K. M. Shen [1,3✉]

The evolution of Sr$_2$IrO$_4$ upon carrier doping has been a subject of intense interest, due to its similarities to the parent cuprates, yet the intrinsic behaviour of Sr$_2$IrO$_4$ upon hole doping remains enigmatic. Here, we synthesize and investigate hole-doped Sr$_{2-x}$K$_x$IrO$_4$ utilizing a combination of reactive oxide molecular-beam epitaxy, substitutional diffusion and in-situ angle-resolved photoemission spectroscopy. Upon hole doping, we observe the formation of a coherent, two-band Fermi surface, consisting of both hole pockets centred at $(\pi, 0)$ and electron pockets centred at $(\pi/2, \pi/2)$. In particular, the strong similarities between the Fermi surface topology and quasiparticle band structure of hole- and electron-doped Sr$_2$IrO$_4$ are striking given the different internal structure of doped electrons versus holes.

[1] Laboratory of Atomic and Solid State Physics, Department of Physics, Cornell University, Ithaca, New York 14853, USA. [2] Department of Materials Science and Engineering, Cornell University, Ithaca, New York 14853, USA. [3] Kavli Institute at Cornell for Nanoscale Science, Ithaca, New York 14853, USA. [4] Present address: Department of Materials Science and Engineering, University of Wisconsin, Madison, Wisconsin 53706, USA. ✉email: kmshen@cornell.edu

The spin–orbit-coupled Mott insulator $Sr_2IrO_4$ exhibits a fascinating interplay between numerous competing energy scales, including spin–orbit coupling (SOC), Coulomb repulsion ($U$), Hund's coupling ($J_H$), and has thus been subject of much recent interest[1–5]. Close similarities between $Sr_2IrO_4$ and the parent cuprate $La_2CuO_4$ have also led to theoretical proposals that hole- and electron-doped $Sr_2IrO_4$ could likewise exhibit unconventional superconductivity[2,6,7]. To date, the majority of work has focused on the electron-doped side of the phase diagram with reports of a momentum-dependent pseudogap in $Sr_{2-x}La_xIrO_4$[8] and a $d$-wave-like gap in surface $K$-doped $Sr_2IrO_4$[4,5]. These are both features associated with cuprates, although no direct signature of superconductivity has been conclusively identified. On the other hand, the behaviour of $Sr_2IrO_4$ upon hole doping is less clear, as the vast majority of studies have examined $Sr_2Ir_{1-x}Rh_xO_4$[9–13] and Rh substitution introduces a number of complexities beyond the doping of holes. As Rh is introduced into the $IrO_2$ planes, it removes a local $J_{eff} = 1/2$ moment, leading to pairs of $Rh^{3+}$ and $Ir^{5+}$ non-magnetic impurities[14], which is analogous to Zn or Ni substitution for Cu in cuprates[15,16]. In cuprate superconductors, the preferred doping sites are those that are not part of the $CuO_2$ planes, e.g., the A-site of cuprates with formula $A_2CuO_4$ ($A$ = Sr or Ba). The A-sites are preferred because of the lower disorder potential that they evoke compared with sites within the $CuO_2$ planes; by analogy with $Sr_2IrO_4$, it is likely similarly advantageous to minimize disorder by doping the Sr-site with an appropriate dopant rather than to dope the Ir-site with Rh. Furthermore, Rh substitution should also change the average strength of the SOC[11], a key ingredient in the formation of the low-energy electronic structure.

To reveal the intrinsic behaviour of $Sr_2IrO_4$ upon hole doping, it is thus desirable to investigate a system without the intertwined complexity caused by Rh substitution. In principle, substitution of a monovalent alkali (e.g., $Na^+$ or $K^+$) on the A-site for divalent $Sr^{2+}$ should result in hole doping without the additional complexities introduced by Rh substitution, similar to the hole doping of the cuprate $Ca_{2-x}Na_xCuO_2Cl_2$ by Na substitution[17–19]. This has been demonstrated in $A_{2-x}K_xIrO_4$ ($A$ = Sr, Ba) to preserve the long-range magnetic order at moderate amount of doping (up to $x = 0.055$)[20–22]. At present, however, no detailed spectroscopic measurements of any kind have been reported for $A_{2-x}K_xIrO_4$ presumably due to the difficulty in synthesizing high-quality bulk single crystals. To overcome this challenge, we employ a combination of reactive oxide molecular-beam epitaxy (MBE) to synthesize initially undoped $Sr_2IrO_4(001)$ thin films, followed by a substitutional diffusion process[23,24], which allows us to substitute K for Sr. This approach circumvents the extremely high vapour pressure of $KO_2$ at typical growth temperatures ($\approx 10^{-2}$ torr at 850 °C)[25], which would otherwise prevent the direct incorporation of K into the thin film; additional details about this process can be found in the Methods section. Afterwards, in-situ angle-resolved photoemission spectroscopy (ARPES) measurements of $Sr_{1.93}K_{0.07}IrO_4$ thin films allow us to disentangle, for the first time, the effects of hole doping from changes in the SOC, magnetic landscape, and strong disorder scattering in the layered iridates. In doing so, we reveal that upon hole doping, coherent quasiparticles emerge together with the collapse of the Mott gap, in contrast to what has previously been reported with Rh substitution.

## Results

### Chemical potential shift with doping
In principle, the addition of K into $Sr_2IrO_4$ can result in either hole or electron doping. If K does not replace Sr, either when adsorbed on the surface[4,5,26] or intercalated, this should result in electron doping. On the other

hand, if $K^+$ substitutes for $Sr^{2+}$, this should result in hole doping. To conclusively demonstrate hole doping, we measured the shift in chemical potential $\Delta\mu$ between undoped, K surface-doped, and K-substituted samples. In Fig. 1a, we show representative energy distribution curves (EDCs) of the valence band from a single $Sr_2IrO_4$ sample when it is (i) initially undoped (black), (ii) following surface K-deposition (green), and finally (iii) after substitutional diffusion of K for Sr (purple). K surface deposition in step (ii) causes a shift of the spectra by $\Delta\mu = +0.5 \pm 0.1$ eV, consistent with electron doping as previously reported by Kim et al.[4,5]. In contrast, following substitutional diffusion in step (iii), all features are shifted to lower binding energy by $\Delta\mu = -0.4 \pm 0.1$ eV, in the direction consistent with hole doping as established by Louat et al.[12] via Rh substitution. This process also results in a clear change in the K $3p$ core levels (Fig. 1b), as K is oxidized and substituted into the SrO layer. We excluded the possibility of hole doping via Sr vacancies[27], interstitial oxygen, or oxygen vacancies[28], by verifying that the post-growth annealing steps had no observable effect when the K-deposition step was omitted (see Supplementary Note 2).

### Evolution of low-energy electronic structure
To investigate the effects of K substitution, in Fig. 2 we compare an isoenergy map of an undoped $Sr_2IrO_4$ film at 0.3 eV binding energy (sample exhibited no weight at $E_F$) with a Fermi surface map of the same sample following K substitution. The isoenergy map of undoped $Sr_2IrO_4$ in Fig. 2a closely resembles those reported for undoped bulk crystals of $Sr_2IrO_4$[31]. In reality, when an electron is removed from $Sr_2IrO_4$ (e.g., by photoemission or hole doping) $5d^4$ holes are introduced in the $IrO_2$ plane, where the low-energy excitations are in fact a non-magnetic singlet $J_{eff} = 0$ and a magnetic triplet state $J_{eff} = 1$ (as described by Pärschke et al.[32]). To remain consistent with the existing iridate literature, these bands may be referred to as $J_{eff} = 1/2$ and $J_{eff} = 3/2$ bands, following the convention for the undoped $5d^5$ configuration. Furthermore, an electron addition $5d^6$ state is non-magnetic with no degrees of freedom, suggesting that electrons and holes may couple differently to the local magnetic environment. The top of the occupied $J_{eff} = 1/2$ band is at $(\pi, 0)$ and $(0, \pi)$, and the top of the $J_{eff} = 3/2$

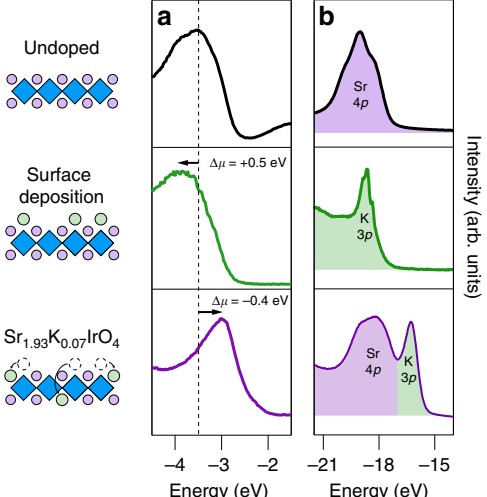

**Fig. 1 Photoemission measurements showing change in chemical potential and core level spectra upon doping $Sr_2IrO_4$. a** Measurement of the chemical potential shift, $\Delta\mu$, for a pristine undoped sample (top), after K surface deposition (middle), and after substitutional diffusion (bottom) with $h\nu = 21.2$ eV at $k_x,k_y = (0,0)$. **b** Corresponding core level spectra measured with He II photons ($h\nu = 40.8$ eV). K peak locations are consistent with reference spectra for elemental[29] and oxidized K[30].

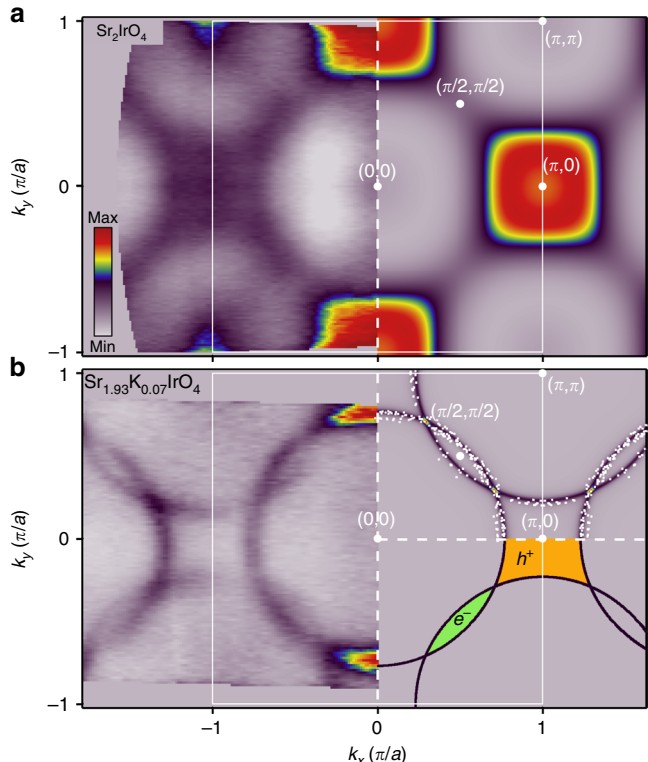

**Fig. 2 Constant energy maps measured with $h\nu = 21.2$ eV and simulated using tight-binding model. a** Energy isosurface at $E_B = 0.3$ eV for undoped $Sr_2IrO_4$ (left) together with a broadened tight binding + spin–orbit coupling + $U$ calculation, with $U = 2$ eV (right). All data are shown in a tetragonal Brillouin zone (1 Ir per unit cell), which ignores back-folding due to the in-plane octahedral rotations, which causes a $\sqrt{2} \times \sqrt{2}$ $R45°$ reconstruction. **b** Fermi surface of $Sr_{1.93}K_{0.07}IrO_4$ showing ARPES data (left, averaged $\pm 20$ meV around $E_F$) together with a tight binding and spin–orbit coupling simulation (right) with $U = 0$ eV; hole and electron pockets are shaded orange and green, respectively. Experimentally extracted $k_F$s are shown as white dots on the tight-binding model simulation.

band lies at (0,0) and ($\pi$, $\pi$), using the notation assuming an idealized, undistorted $IrO_2$ square lattice, as has been customary in the literature. Upon hole doping, the Fermi surface of $Sr_{2-x}K_xIrO_4$ is now clearly composed of small, elliptical electron pockets closed about ($\pi/2$, $\pi/2$) and larger square hole pockets centred around ($\pi$, 0) and (0, $\pi$). A measurement of the Luttinger volumes of these pockets indicates a hole doping of $x = 0.07 \pm 0.02$, consistent with the observed shift in chemical potential, conclusively demonstrating hole doping via K substitution.

In Fig. 3a-d, we compare spectra from undoped $Sr_2IrO_4$ and $Sr_{2-x}K_xIrO_4$ along both the (0,0)-(0,$\pi$) and ($\pi/2$,0)-($\pi/2$,$\pi$) momentum cuts. For undoped $Sr_2IrO_4$, the broad but distinct $J_{eff} = 1/2$ band reaches its maximum at (0,-$\pi$) as reported previously in bulk single crystals, whereas for $Sr_{1.93}K_{0.07}IrO_4$ samples, this band evolves into a sharp, well-defined quasiparticle band. Comparisons of the individual EDCs at $k = (0, 1.2\pi)$ from the doped and undoped samples are shown in Fig. 3e), showing that the broad $J_{eff} = 1/2$ excitations in $Sr_2IrO_4$ evolve into a well-defined quasiparticle peak for $Sr_{1.93}K_{0.07}IrO_4$. In Fig. 3b, d, we compare spectra from $Sr_2IrO_4$ and $Sr_{1.93}K_{0.07}IrO_4$ around ($\pi/2$, $\pi/2$), which in the doped compound intersects the small, elliptical electron pockets as is clear from the momentum distribution curve (MDC) at $E_F$ (Fig. 3d). Corresponding EDCs from $Sr_2IrO_4$ and $Sr_{1.93}K_{0.07}IrO_4$ taken at $k = (\pi/2, 0.6\pi)$ are shown in Fig. 3e. The lack of a large, uniform pseudogap (we observe a leading

edge midpoint of less than 5 meV—see Supplementary Note 3—which is substantially less than the 30 meV shift reported in the Rh-doped samples[12,13]), and the presence of quasiparticle peaks, which are absent in $Sr_2Ir_{1-x}Rh_xO_4$, are both consistent with lower disorder scattering, in the absence of substitutional disorder directly in the $IrO_2$ plane.

**Tight-binding model.** A key distinction between $Sr_{1.93}K_{0.07}IrO_4$ and earlier studies of $Sr_2Ir_{1-x}Rh_xO_4$ is the clear presence of elliptical electron pockets, shown both in the Fermi surface map in Fig. 2b and in the band dispersion in Fig. 3d, which were not observed in $Sr_2Ir_{1-x}Rh_xO_4$. To better understand the origin of these features, we employ a tight-binding parametrization of the $t_{2g}$ bands following refs. [8,33,34], which has previously shown good agreement with photoemission data[8]:

$$\mathcal{H} = \sum_{\langle ij \rangle \alpha\beta\sigma} t_{ij}^{\alpha\beta} c_{i\alpha\sigma}^\dagger c_{j\beta\sigma} + \sum_{i,\alpha=d_{xy}} \Delta_t c_{i\alpha\sigma}^\dagger c_{i\alpha\sigma} + \lambda \sum_i \overrightarrow{L}_i \cdot \overrightarrow{S}_i, \quad (1)$$

where $\langle ij \rangle$ are nearest-neighbour pairs of Ir sites, $\alpha$ and $\beta$ index the $t_{2g}$ orbitals, $t_0 = 0.35$ eV, $\sigma$ indicates the spin, $\Delta t = 0.15$ eV is the tetragonal crystal field splitting and $\lambda = 0.57$ eV is the SOC parameter. These are the same values used in ref. [8]. The Coulomb repulsion $U$ is implemented as an additional self-consistent mean-field term, which is proportional to the average electron density of each band. Additional details of the calculation can be found in Supplementary Note 4.

In Fig. 4, we show the tight-binding band structure together with extracted experimental dispersions from both $Sr_2IrO_4$ and $Sr_{1.93}K_{0.07}IrO_4$. We find good agreement in both the isoenergy maps (Fig. 2a) and extracted band dispersions (Fig. 4a) for the undoped case for a value of $U = 2$ eV, consistent with earlier studies of undoped $Sr_2IrO_4$[8]. Rigidly shifting $\mu$ into the top of the $J_{eff} = 1/2$ band of this band structure would result in a Fermi surface comprised solely of hole pockets centred at ($\pi$, 0) and (0, $\pi$), as shown in Fig. 4b. This is reminiscent of the Fermi surface of $Sr_2Ir_{1-x}Rh_xO_4$, where it was argued that the Mott gap is largely preserved up to a hole doping of $x = 0.20$[9,10,12,13], but counter to our observations in $Sr_{1.93}K_{0.07}IrO_4$.

A few important distinctions can be made between 7% hole-doped $Sr_{2-x}K_xIrO_4$ and $Sr_2Ir_{0.93}Rh_{0.07}O_4$ with a comparable hole doping from ref. [12]. First, the top of the $J_{eff} = 1/2$ band at ($\pi$,0) differs by approximately 0.1 eV (approximately +0.04 eV above $E_F$ for Rh-doped, approximately +0.15 above $E_F$ for K-doped). This demonstrates that $Sr_2Ir_{0.93}Rh_{0.07}O_4$ is consistent with the rigid band shift scenario in Fig. 4a. Although $Sr_2Ir_{0.93}Rh_{0.07}O_4$ only shows appreciable spectral weight near $E_F$ around ($\pi$,0), albeit without sharp spectral features, $Sr_{1.93}K_{0.07}IrO_4$ possesses a two-sheet Fermi surface comprised of sharp, well-defined quasiparticle bands. In particular, the two-pocket fermiology observed in $Sr_{1.93}K_{0.07}IrO_4$ strongly suggests a scenario where the Mott gap has collapsed when $U$ becomes sufficiently small (Fig. 4d-f), as the elliptical electron pocket originates from the upper Hubbard band itself. Although our data stands in clear contrast to studies of $Sr_2Ir_{1-x}Rh_xO_4$, it bears qualitative resemblance to the case of electron doping in $Sr_{2-x}La_xIrO_4$, where both hole and electron pockets have likewise been reported[8]. Nevertheless, electron vs. hole doping can be clearly distinguished from the relative sizes of the hole and electron pockets, as shown in Fig. 4e, f.

This striking similarity between electron-doped $Sr_{2-x}La_xIrO_4$ and hole-doped $Sr_{2-x}K_xIrO_4$ in the global quasiparticle band structure and Fermi surface topology, apart from a shift in the chemical potential, is unexpected given the differences in the internal structure of the doped electrons (Ir $5d^6$ in a simple $J_{eff} = 0$ state) vs. doped holes (Ir $5d^4$ with a complex spin–orbit-coupled

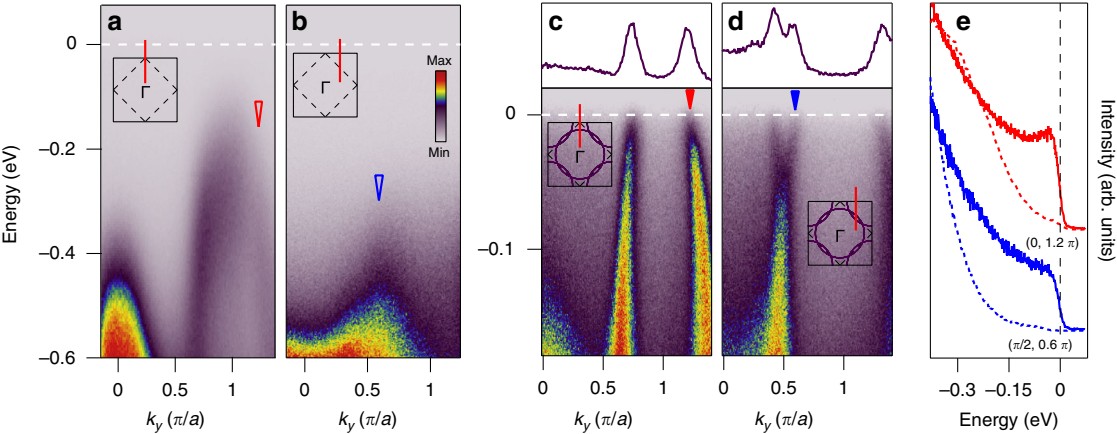

**Fig. 3 Dispersions measured with $h\nu = 21.2$ eV. a** ARPES spectra along $(0,0)-(0,\pi)$ and **b** along $(\pi/2,0)-(\pi/2,\pi)$ in undoped Sr$_2$IrO$_4$; insets show the Brillouin zone with red lines indicating the direction of the dispersion. Blue and red arrows show the momentum of the EDCs shown in **e. c, d** Corresponding ARPES spectra for Sr$_{1.93}$K$_{0.07}$IrO$_4$ intersecting the square hole pocket **c** and elliptical electron pocket **d** with MDCs at $E_F$ shown at the top. **e** EDCs at $k_F$ in the doped samples (solid lines) and corresponding EDCs at the same $k$ in the undoped samples (dashed lines), indicating a clear shift of spectral weight towards the Fermi level and a quasiparticle peak at $(0, 1.2\pi/a)$.

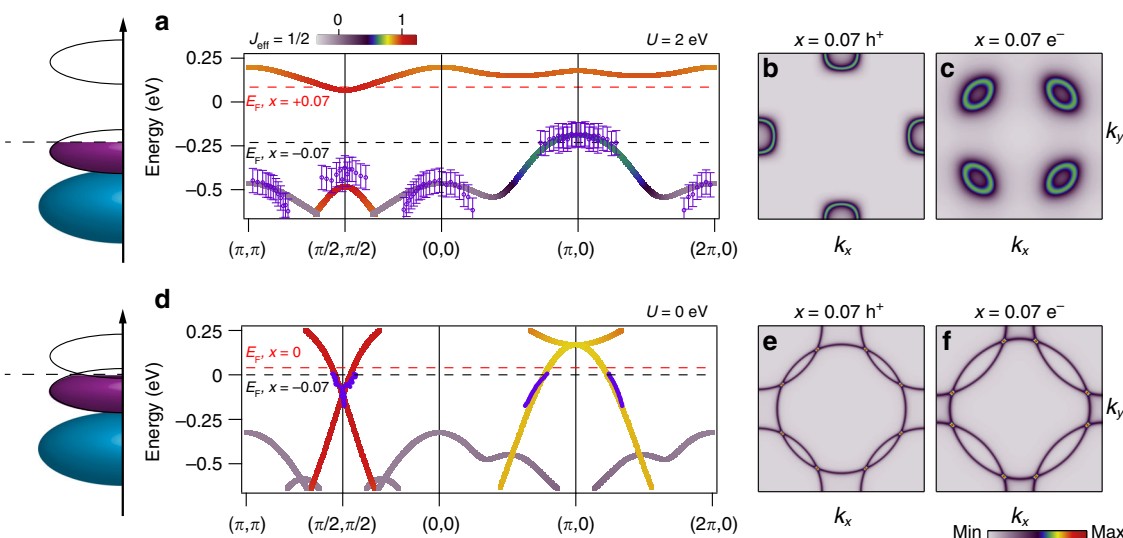

**Fig. 4 Tight-binding model compared with extracted band dispersions. a** Tight-binding model of the band structure with an additional mean-field Coulomb repulsion term $U = 2$ eV illustrating the behaviour of undoped Sr$_2$IrO$_4$ with extracted experimental dispersions of undoped Sr$_2$IrO$_4$ shown (purple circles), together with a schematic density of states, error bars indicated estimated uncertainty due to broad bands characteristic of insulating Sr$_2$IrO$_4$. Dashed black line indicates the chemical potential with $x = 0.07$ hole doping in a rigid band shift scenario similar to Rh-doped Sr$_2$IrO$_4$, dashed red line indicates chemical potential with $x = 0.07$ electron doping. Simulated tight-binding Fermi surfaces in a rigid band shift scenario are shown for **b** hole and **c** electron doping. **d** Tight-binding model with $U = 0$ eV, where the Mott gap has collapsed with extracted experimental dispersions of Sr$_{1.93}$K$_{0.07}$IrO$_4$ (purple circles) together with a schematic density of states. Simulated tight-binding Fermi surfaces when $U = 0$ eV shown for hole **e** and electron doping **f**.

multiplet structure). This surprising apparent symmetry between electron and hole doping should motivate future many-body calculations (e.g., Hubbard, $t$–$J$ model, or dynamical mean-field theory calculations), which explicitly consider the complex multiplet structure of hole-doped Sr$_2$IrO$_4$.

Despite the apparent symmetry of the global electronic structure upon both electron and hole doping, there remain important distinctions between the two systems at the lowest energy scales. Whereas the electron-doped iridates (surface K or La substitution) in a similar doping range exhibit a large (20 meV), *d*-wave-like pseudogap at $E_F$, we do not experimentally resolve a pseudogap to within 5 meV.

## Discussion

The differences between Sr$_{2-x}$K$_x$IrO$_4$ and Sr$_2$Ir$_{1-x}$Rh$_x$O$_4$ allow us to elucidate the intrinsic effects of hole doping vs. the additional effects caused by Rh substitution. The lack of electron pockets in the Rh-doped materials is suggestive of a rigid band shift scenario where the Mott gap is largely preserved, in contrast to Sr$_{2-x}$K$_x$IrO$_4$, where we find that the Mott gap collapses and coherent quasiparticle excitations are formed. A possible explanation is that structural and magnetic disorder in the IrO$_2$ planes may cause holes to be strongly localized around Rh dopants, inhibiting free carriers from effectively screening the Mott gap, whereas for the K-doped materials, the carriers are

more delocalized and are thereby able to more efficiently screen the strong Coulomb interactions. In addition, the existence of coherent quasiparticle peaks and the lack of a large pseudogap in $Sr_{2-x}K_xIrO_4$ also suggests that the incoherent metallic and the large pseudogap reported in $Sr_2Ir_{1-x}Rh_xO_4$ are likely induced by substitutional disorder in the $IrO_2$ plane[12], rather than an intrinsic property of hole-doped iridates.

Our findings point towards a universal underlying electronic structure upon doping $Sr_2IrO_4$, irrespective of the sign of the carriers, and thus a more symmetric doping phase diagram than previously realized. This stands in contrast to the cuprates, which exhibit a fundamental asymmetry between electron and hole doping[35]. Whereas the evolution of the Fermi surface contours in electron-doped cuprates can be qualitatively modelled by introducing a $(\pi, \pi)$ spin–density wave via a conventional weak coupling phenomenology[36], explaining the disconnected Fermi arcs of the hole-doped cuprates remains an outstanding challenge for sophisticated many-body approaches, which necessarily include strong local interactions[37]. Furthermore, the Mott gap in the cuprates appears far more robust upon doping, where spectral weight is gradually transferred from the Hubbard to low-energy quasiparticle bands[38,39], whereas the gap in both hole- and electron-doped iridates appears to collapse far more rapidly. These differences may arise from the fundamentally weaker on-site Coulomb repulsion in the Ir $5d$ orbitals vs. the Cu $3d$ orbitals. Another distinction is that the cuprates are charge-transfer insulators where the hole- and electron-doped states have stronger O $2p$ vs. Cu $3d$ character, respectively, whereas the iridates are better described as Mott insulators where both the hole- and electron-doped states are of primarily Ir $5d$ orbital character, although they have very different internal magnetic structure[32]. Future work including detailed doping dependence of the electronic structure and magnetism, and a study of the electronic structure with advanced many-body techniques such as dynamical mean-field theory used in electron-doped $Sr_2IrO_4$[40,41], will be necessary to fully explain the collapse of the Mott gap in hole-doped $Sr_{2-x}K_xIrO_4$ and the symmetric, universal electronic structure upon both hole and electron doping.

## Methods

**Film growth**. Epitaxial $Sr_2IrO_4(001)$ thin films were grown on single-crystalline $(LaAlO_3)_{0.3}$ $(SrAl_{1/2}Ta_{1/2}O_3)_{0.7}$ substrates by MBE at a substrate temperature of 850 °C as measured by an optical pyrometer with a measurement wavelength of 980 nm, in a background partial pressure of $1 \times 10^{-6}$ torr of distilled ozone (80% $O_3 + 20\%$ $O_2$). A 7 nm buffer layer of $SrIrO_3$ was initially deposited, followed by 20 nm of $Sr_2IrO_4$; the conducting $SrIrO_3$ layer facilitated measurements below 100 K. Additional details about the growth may be found in refs. [42,43] and in Supplementary Note 1.

**Substitutional diffusion**. K substitution was achieved through a substitutional diffusion process[23,24] where K was deposited on an undoped $Sr_2IrO_4$ film using a SAES evaporator at temperatures below 70 K and then annealed at a temperature of 300 °C in vacuum, followed by exposure to $1 \times 10^{-6}$ torr of ozone at 300 °C for 20 min (both 10% and 80% $O_3$ were used). Multiple doped samples synthesized and investigated in this study showed highly consistent values in the chemical potential shift, near-$E_F$ electronic structure, and extracted hole concentrations, despite significant variations in the amounts of K deposited, annealing times, or ozone concentration. This method was used because of the extremely high vapour pressure of $KO_2$ ($\approx 1 \times 10^{-2}$ torr at 850 °C), which prevents the direct incorporation of K into a $Sr_{2-x}K_xIrO_4$ film at its growth temperature of 850 °C.

**ARPES measurements**. Following growth, the samples were transferred for ARPES measurements using He I$\alpha$ ($h\nu = 21.2$ eV) photons with an energy resolution of $\Delta E = 11$ meV at a temperature of 15 K. All stages described occurred within a single ultrahigh vacuum manifold, i.e., the samples were never exposed to air.

## Data availability
The data that support the findings of this study are available from the corresponding author upon reasonable request.

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

## Acknowledgements

This work was supported through the National Science Foundation [Platform for the Accelerated Realization, Analysis, and Discovery of Interface Materials (PARADIM)] under Cooperative Agreement Number DMR-1539918, NSF DMR-1709255, and the Air Force Office of Scientific Research Grant Number FA9550-15-1-0474. This research is funded in part by the Gordon and Betty Moore Foundation's EPiQS Initiative through Grant Number GBMF3850 to Cornell University. J.N.N. and B.D.F. acknowledge support from the NSF Graduate Research Fellowship under Grant Number DGE-1650441. C.T.P. acknowledges support from the Center for Bright Beams, NSF award PHY-1549132.

Substrate preparation was performed in part at the Cornell NanoScale Facility, a member of the National Nanotechnology Coordinated Infrastructure (NNCI), which is supported by the NSF (Grant Number ECCS-1542081). This work made use of the Cornell Center for Materials Research Shared Facilities, which are supported through the NSF MRSEC programme (DMR-1719875).

## Author contributions

J.N.N. performed the ARPES measurements and synthesized the thin films, with assistance from C.T.P., B.D.F., and J.K.K., and guidance from D.G.S. and K.M.S. J.N.N. performed the tight-binding calculations. J.N.N. and K.M.S. conceived the project and wrote the manuscript with contributions from all authors. K.M.S. supervised the project.

## Competing interests

The authors declare no competing interests.
