## [Peer Review File · Nature Communications]

Reviewers' Comments:

Reviewer #1:

Remarks to the Author:

The paper by JN Nelson et al. present very interesting ARPES results of the “genuinely” hole-doped iridate — $\text{Sr}_{2-x}\text{K}_x\text{IrO}_4$.

In my opinion, the main finding of the paper is that “upon hole doping, coherent quasiparticles emerge together

with the collapse of the Mott gap”. This is a novel result, since all previous hole-doped iridate studies (which probably

must have introduced also other changes to the undoped iridium oxide structure) did not report a transition

to a system which can be regarded as almost noninteracting or at most weakly interacting. That is why I think that such an

interesting study should be published in Nature Communications. Nevertheless, before I can fully recommend

the manuscript for publication I would like the Authors to address the following three issues:

(1) Interpretation of the undoped ARPES data:

The Authors denote the dominant features of the ARPES spectrum of Sr_2IrO_4 as $J_{\text{eff}}=1/2$ and $J_{\text{eff}}=3/2$ bands. While such a description

is quite common in the iridate literature, in my opinion this is not the correct way of describing this correlated spectrum: in “reality”

what happens in the ARPES experiment is that one introduces a d^4 configuration (“ d^4 hole”) in the Sr_2IrO_4 plane — and the

eigenstates of the latter should be denoted as having $J_{\text{eff}}=0$, $J_{\text{eff}}=1$ quantum numbers (these are the 4 lowest lying “spin-orbit coupled”

multiplets of the 15 d^4 ion multiplets). Thus, the proper labelling of the various features of the ARPES spectrum of Sr_2IrO_4 spectrum (of the “bands”, although

this is also not a fully correct description in the correlated limit) should bear the $J_{\text{eff}}=0$, $J_{\text{eff}}=1$, etc. quantum numbers,

cf. EM Paerschke et al., Nature Communications 8, 686 (2017) for details.

(2) Presentation of the model results with finite U:

I understand that the result shown in Fig. 4(a-c) is obtained using the mean-field calculations — this should be stated explicitly in the figure caption.

(3) Similarities between the electron and hole doped iridates [this is actually the main point of the report]:

While I appreciate that the Authors would like to discuss this very important problem, I think that the discussion might be better organised and a few issues could (or maybe even should) be clarified:

(a) Experimental signatures of the similarities between the electron- and hole-doped iridates:

In my opinion, the respective ARPES spectra are still quite different: (i) as the Authors mention “there exist subtle differences such

as the antinodal pseudogap in $\text{Sr}_{1.9}\text{La}_{0.1}\text{IrO}_4$, which has been studied both experimentally with ARPES [8]”;

(ii) also the ARPES studies on the electron-doped iridates reported by Kim et al. [References [4-5]

in the paper]

suggest clear signatures of the Fermi arcs and the important role played by correlations in the electron-doped iridates.

That is why I would not support the claim that the electron and hole-doped iridates are similar and I would appreciate, if the Authors can revise the relevant discussion of that issue in the paper accordingly (including, but not limited to, the abstract).

(b) Theoretical situation:

In my opinion, on the theory side, the physics of the electron- and hole-doped iridates should be entirely different, at least, if we assume that the correlated physics plays any significant role in the system:

In the electron-doped case one introduces "genuine" holes into the system (d^6 configuration on iridium ions; as a result of large splitting to the e_g 's this can be regarded a $J_{\text{eff}} = 0$ state, i.e. this is just a single "hole" or "electron" as known from the "single-band" cuprate studies). Thus, on the theoretical side the problem can be modelled by a single-band Hubbard or t - J —like model and in the "underdoped" regime one expects a "typical" "correlated" ARPES spectrum, e.g. showing the pseudogaps. Indeed, this is what the experiments on the electron-doped iridates suggest [see point (2)(a)].

On the other hand, in the hole-doped iridates locally one introduces the d^4 states [see discussion in (1)] with the 15 possible multiplet states per site (which then can "disperse" and form the resulting "bands" in the ARPES spectrum). Of course, it is not clear how these 15 multiplets "conspire" to form the observed almost noninteracting bands, i.e. why in the end of the day it looks as upon doping the whole system can be described as almost noninteracting already at 7% doping, as the Authors report. In my point of view, this the most interesting question that is left "open" from this study — this requires studying a doped multiband Hubbard / t - J like model with large spin-orbit coupling (a very complicated study, of course...).

While I understand that the above discussion might be seen as a bit subjective and rather lengthy, I would think that at least some of it might be instructive to the Readers — especially the one regarding the need for further *multiband* Hubbard / t - J studies with large spin-orbit coupling.

Reviewer #2:

Remarks to the Author:

The authors report ARPES study of hole-doped $\text{Sr}_{2-x}\text{K}_x\text{IrO}_4$ grown by MBE. Sr_2IrO_4 has received much attention due to its similarity with cuprate superconductors in terms of the electronic evolution as carriers are doped into the parent Mott insulator. For iridates, many studies are focused on the electron-doped side, partly due to the fact that hole doping is usually done by

substitution of Ir ions by Rh, a process that generates disorder in the IrO₂ plane and thus can involve unwanted side effects. To my knowledge, this is a first high quality ARPES study on a hole doped iridate though substitution of Sr by K, which is more amenable to studying intrinsic electronic evolution with hole doping. The authors have made very high quality samples as judged by the quality of their ARPES data. The data are impressive and I believe that much insight can be gained by studying the doping evolution systematically.

That being said, the manuscript only provides data on one doped sample, which is not enough to draw any important conclusion. The authors study 7% doped sample and find that the Mott gap has collapsed. As the authors note, Fermi liquid phase is reached at a relatively low doping level in iridates compared to cuprates, and so authors might have missed the interesting evolution that takes place at lower doping levels. In fact, in the electron doped side, a metal phase with LDA-like Fermi surface is observed already at ~8% doping. I think this work will have much more impact if the authors can grow one or two more intermediate doping levels samples and provide ARPES data on them. I understand that MBE growth of iridate films is a very difficult task, but as is written now the manuscript provides very limited information and thus is not sufficient for publication in Nature communications.

Response to Reviewer #1:

"The paper by JN Nelson et al. present very interesting ARPES results of the "genuinely" hole-doped iridate — $\text{Sr}_{2-x}\text{K}_x\text{IrO}_4$. In my opinion, the main finding of the paper is that "upon hole doping, coherent quasiparticles emerge together with the collapse of the Mott gap". This is a novel result, since all previous hole-doped iridate studies (which probably must have introduced also other changes to the undoped iridium oxide structure) did not report a transition to a system which can be regarded as almost noninteracting or at most weakly interacting. That is why I think that such an interesting study should be published in Nature Communications. Nevertheless, before I can fully recommend the manuscript for publication I would like the Authors to address the following three issues:"

We thank the reviewer #1 for their comments regarding our work presenting the first spectroscopic study of $\text{Sr}_{2-x}\text{K}_x\text{IrO}_4$ and their positive response. We address reviewer #1's comments below:

" (1) Interpretation of the undoped ARPES data:

The Authors denote the dominant features of the ARPES spectrum of Sr_2IrO_4 as $J_{\text{eff}}=1/2$ and $J_{\text{eff}}=3/2$ bands. While such a description is quite common in the iridate literature, in my opinion this is not the correct way of describing this correlated spectrum: in "reality" what happens in the ARPES experiment is that one introduces a d^4 configuration (" d^4 hole") in the Sr_2IrO_4 plane — and the eigenstates of the latter should be denoted as having $J_{\text{eff}}=0$, $J_{\text{eff}}=1$ quantum numbers (these are the 4 lowest lying "spin-orbit coupled" multiplets of the 15 d^4 ion multiplets). Thus, the proper labelling of the various features of the ARPES spectrum of Sr_2IrO_4 spectrum (of the "bands", although this is also not a fully correct description in the correlated limit) should bear the $J_{\text{eff}}=0$, $J_{\text{eff}}=1$, etc. quantum numbers, cf. EM Paerschke et al., Nature Communications 8, 686 (2017) for details."

We thank Reviewer #1 for bringing up this issue and agree with them that indeed the proper labeling for the undoped and hole doped lowest energy excitations are $J_{\text{eff}} = 0$ and $J_{\text{eff}} = 1$. We appreciate their suggestion that using this notation helps to illustrate the important differences between electron and hole doping which are rather unique to Sr_2IrO_4 and have modified our manuscript as follows :

“ In reality, when an electron is removed from Sr_2IrO_4 (e.g. by photoemission or hole doping) $5d^4$ holes are introduced in the IrO_2 plane, where the low energy excitations are in fact a nonmagnetic singlet $J_{\text{eff}}=0$ and a magnetic triplet state $J_{\text{eff}}=1$ (as described by Pärshcke et al.³³ [Parschke *et al. Nat. Comm.* **8**, 686 (2017)]). To remain consistent with the existing iridate literature, these bands may be referred to as $J_{\text{eff}}=1/2$ and $J_{\text{eff}}=3/2$ bands, following the convention for the undoped $5d^5$ configuration. Furthermore, an electron addition $5d^6$ state is non magnetic with no degrees of freedom, suggesting that electrons and holes may couple differently to the local magnetic environment.” (lines 111-121).

“(2) Presentation of the model results with finite U:

I understand that the result shown in Fig. 4(a-c) is obtained using the mean-field calculations — this should be stated explicitly in the figure caption.”

We thank Reviewer #1 for this comment and have amended the figure caption, for additional clarity we also state in lines 168-170 “The Coulomb repulsion U is implemented as an additional self-consistent mean-field term, which is proportional to the average electron density of each band.”

“(3) Similarities between the electron and hole doped iridates [this is actually the main point of the report]. While I appreciate that the Authors would like to discuss this very important problem, I think that the discussion might be better organised and a few issues could (or maybe even should) be clarified:

(a) Experimental signatures of the similarities between the electron- and hole-doped iridates:

In my opinion, the respective ARPES spectra are still quite different: (i) as the Authors mention “there exist subtle differences such as the antinodal pseudogap in $\text{Sr}_{1.9}\text{La}_{0.1}\text{IrO}_4$, which has been studied both experimentally with ARPES [8]”; (ii) also the ARPES studies on the electron-doped iridates reported by Kim et al. [References [4-5] in the paper] suggest clear signatures of the Fermi arcs and the important role played by correlations in the electron-doped iridates. That is why I would not support the claim that the electron and hole-doped iridates are similar and I would appreciate, if the Authors can revise the relevant discussion of that issue in the paper accordingly (including, but not limited to, the abstract).

(b) Theoretical situation:

In my opinion, on the theory side, the physics of the electron- and hole-doped iridates should be entirely different, at least, if we assume that the correlated physics plays any significant role in the system:

In the electron-doped case one introduces “genuine” holes into the system (d^6 configuration on iridium ions; as a result of large splitting to the e_g 's this can be regarded a $J_{\text{eff}}=0$ state, i.e.

this is just a single “hole” or “electron” as known from the “single-band” cuprate studies). Thus, on the theoretical side the problem can be modelled by a single-band Hubbard or t-J—like model and in the “underdoped” regime one expects a “typical” “correlated” ARPES spectrum, e.g. showing the pseudogaps. Indeed, this is what the experiments on the electron-doped iridates suggest [see point (2)(a)].

On the other hand, in the hole-doped iridates locally one introduces the d^4 states [see discussion in (1)] with the 15 possible multiplet states per site (which then can “disperse” and form the resulting “bands” in the ARPES spectrum). Of course, it is not clear how these 15 multiplets “conspire” to form the observed almost noninteracting bands, i.e. why in the end of the day it looks as upon doping the whole system can be described as almost noninteracting already at 7% doping, as the Authors report.

In my point of view, this the most interesting question that is left “open” from this study — this requires studying a doped multiband Hubbard / t-J like model with large spin-orbit coupling (a very complicated study, of course...).

*While I understand that the above discussion might be seen as a bit subjective and rather lengthy, I would think that at least some of it might be instructive to the Readers — especially the one regarding the need for further *multiband* Hubbard / t-J studies with large spin-orbit coupling.”*

We thank Reviewer #1 for these insightful comments and agree that there remain important differences between electron and hole doped iridates which could be more effectively highlighted in our paper. We also agree that a multiband Hubbard or t-J study would greatly help in understanding this system and hope that our work prompts future theoretical studies. We have made numerous changes to the manuscript to address these concerns, including :

Modifying the abstract:

“In particular, the strong similarities between the Fermi surface topology and quasiparticle band structure of hole- and electron-doped Sr_2IrO_4 are striking given the different internal structure of doped electrons versus holes.” (lines 11-13).

Modifying the main text:

“This striking similarity between electron-doped $\text{Sr}_{2-x}\text{La}_x\text{IrO}_4$ and hole-doped $\text{Sr}_{2-x}\text{K}_x\text{IrO}_4$ in the global quasiparticle band structure and Fermi surface topology, apart from a shift in the chemical potential, is unexpected given the differences in the internal structure of the doped electrons (Ir $5d^6$ in a simple $J_{\text{eff}} = 0$ state) versus doped holes (Ir $5d^4$ with a complex spin-orbit coupled multiplet structure). This surprising apparent symmetry between electron and hole doping should motivate future many-body calculations (e.g. Hubbard, t-J model, or dynamical mean field theory calculations) which explicitly consider the complex multiplet structure of hole doped Sr_2IrO_4 .” (lines 207-218).

“Despite the apparent symmetry of the global electronic structure upon both electron and hole doping, there remain important distinctions between the two systems at the lowest energy scales. Whereas the electron doped iridates (surface K or La substitution) in a similar doping range exhibit a large (~ 20 meV), d-wave-like pseudogap at E_F , we do not experimentally resolve a pseudogap to within 5 meV.” (lines 219-225).

Response to Reviewer #2:

“The authors report ARPES study of hole-doped $\text{Sr}_{2-x}\text{K}_x\text{IrO}_4$ grown by MBE. Sr_2IrO_4 has received much attention due to its similarity with cuprate superconductors in terms of the electronic evolution as carriers are doped into the parent Mott insulator. For iridates, many studies are focused on the electron-doped side, partly due to the fact that hole doping is usually done by substitution of Ir ions by Rh, a process that generates disorder in the IrO_2 plane and thus can involve unwanted side effects. To my knowledge, this is a first high quality ARPES study on a hole doped iridate though substitution of Sr by K, which is more amenable to studying intrinsic electronic evolution with hole doping. The authors have made very high quality samples as judged by the quality of their ARPES data. The data are impressive and I believe that much insight can be gained by studying the doping evolution systematically.

That being said, the manuscript only provides data on one doped sample, which is not enough to draw any important conclusion. The authors study 7% doped sample and find that the Mott gap has collapsed. As the authors note, Fermi liquid phase is reached at a relatively low doping level in iridates compared to cuprates, and so authors might have missed the interesting evolution that takes place at lower doping levels. In fact, in the electron doped side, a metal phase with LDA-like Fermi surface is observed already at ~8% doping. I think this work will have much more impact if the authors can grow one or two more intermediate doping levels samples and provide ARPES data on them. I understand that MBE growth of iridate films is a very difficult task, but as is written now the manuscript provides very limited information and thus is not sufficient for publication in Nature communications.”

We thank Reviewer #2 for their positive comments about our manuscript and agree that substitutionally doped $\text{Sr}_{2-x}\text{K}_x\text{IrO}_4$ is an ideal platform for studying the evolution of the electronic structure with hole doping. We absolutely agree that a detailed doping dependence of the electronic structure is an important and natural next step in the study of hole doped iridates. Along these lines, we have indeed attempted to synthesize $\text{Sr}_{2-x}\text{K}_x\text{IrO}_4$ with different levels of hole doping, but have been unable to achieve hole doping levels appreciably different from 7% (as will be described below). Nevertheless, we believe that our results provide very important insights into the hole doping evolution of the iridates, particularly when contrasted to Sr_2IrO_4 doped with similar concentrations of La and Rh, and warrant publication in Nature Communications.

To underscore the important differences between our “intrinsically” hole doped $\text{Sr}_{2-x}\text{K}_x\text{IrO}_4$ and Rh-doped $\text{Sr}_2\text{Ir}_{1-x}\text{Rh}_x\text{O}_4$, we compare our 7% doped sample to a $\text{Sr}_2\text{Ir}_{0.96}\text{Rh}_{0.04}\text{O}_4$ sample from Louat *et al. PRB* **97**, 161109(R) (2018) with a similar hole doping (8% - in this doping range, Cao et al. have shown that each Rh introduces two holes into the IrO_2 planes). To address this more explicitly, we have added the following text to the manuscript :

“ A few important distinctions can be made between 7% hole doped $\text{Sr}_{2-x}\text{K}_x\text{IrO}_4$ and $\text{Sr}_2\text{Ir}_{0.96}\text{Rh}_{0.04}\text{O}_4$ with a comparable hole doping ($x=0.04$ results in 8% holes per Ir) from Ref.¹² [Louat *et al. PRB* **97**, 161109(R) (2018)]. First, the top of the $J_{\text{eff}}=1/2$ band at $(\pi,0)$ differs by approximately 0.25 eV (-0.1 eV below E_F for Rh-doped, approximately +0.15 above E_F for K-doped) suggesting that the efficiency of each doped hole in $\text{Sr}_2\text{Ir}_{1-x}\text{Rh}_x\text{O}_4$ is lower than in our model. While $\text{Sr}_2\text{Ir}_{0.96}\text{Rh}_{0.04}\text{O}_4$ only shows appreciable spectral weight near E_F around $(\pi,0)$, albeit without sharp spectral features, $\text{Sr}_{1.93}\text{K}_{0.07}\text{IrO}_4$ possesses a two-sheet Fermi surface comprised of sharp, well-defined quasiparticle bands. In particular, the two-pocket fermiology

observed in $\text{Sr}_{1.93}\text{K}_{0.07}\text{IrO}_4$ strongly suggests a scenario where the Mott gap has collapsed when U becomes sufficiently small (Figs. 4d-f), since the elliptical electron pocket originates from the upper Hubbard band itself.” (lines 185-200).

Likewise, we can also make comparisons between $\text{Sr}_{1.93}\text{K}_{0.07}\text{IrO}_4$ and electron-doped Sr_2IrO_4 with similar electron carrier concentrations. For example, Kim et al. have shown (by surface K dosing), that at a surface electron doping of 6-8%, a d-wave-like gap is observed. Similarly, in bulk electron-doped $\text{Sr}_{2-x}\text{La}_x\text{IrO}_4$, both de la Torre et al. (*PRL* **115**, 2-6 (2015)) and Terashima et al. (*PRB* **96**, 041106(R) (2017)) have reported a momentum-dependent pseudogap of approximately 20 meV in the range of 5-10% electron doping, in clear contrast to our observations. To address this point, we have also added new text between lines 207 and 225.

We agree with Reviewer #2 that investigating a sequence of different doping levels would certainly be desirable. We have invested significant efforts into synthesizing samples with varying degrees of K substitution, but have been unable to appreciably change the hole concentration. In total, we have synthesized eleven $\text{Sr}_{2-x}\text{K}_x\text{IrO}_4$ thin films for this study. Of these eleven samples, the post-growth K substitutional diffusion conditions have been varied significantly, including (1) the amount of K deposited on the surface has been varied by a factor of 40; (2) the concentration of ozone used in the annealing step (both 10% and 80%); (3) and the amount of time during the vacuum annealing (24 to 40 minutes) and ozone annealing (20 to 55 minutes). Despite varying all these conditions, we have found that of the seven samples that yielded high quality ARPES spectra which can be reliably analyzed, all seven samples give the same extracted hole doping concentration of $7 \pm 2\%$ (from Luttinger volume) and a chemical potential shift of $\Delta\mu = -0.4 \pm 0.1$ eV. We believe that a 7% doping may be energetically favored in the substitutional diffusion process, or that this represents the solubility limit of K in Sr_2IrO_4 . We are continuing to investigate this substitutional diffusion technique, but controlling the doping level is far more challenging than by conventional co-doping or bulk chemical substitution.

To address this issue, we have added a short note in the Methods section on line 294, stating that :

“Multiple doped samples synthesized and investigated in this study showed highly consistent values in the chemical potential shift, near- E_F electronic structure, and extracted hole concentrations, despite significant variations in the amounts of K deposited, annealing times, or ozone concentration.”

We have also added additional detail in the Supplemental Information (lines 31-40) detailing the number of samples synthesized, investigated by ARPES, and the variation in the range of K deposited and annealing conditions.

Despite the fact that we have not been able to appreciably change the hole doping level, we believe that this work represents an important advance in our understanding of doped iridates worthy of publication in Nature Communications. In particular, by making direct comparisons to both Rh-doped and La-doped iridates with similar electron / hole concentrations, we are able to point out important differences between these cases (namely, the collapse of the Mott gap and the absence of a large pseudogap), distinctions which we highlight in this revised manuscript. We believe that these surprising observations will ignite new experimental and theoretical efforts into understanding the physics of electron- and hole-doped iridates.

Correction to paper:

In the original version of the paper we included spectra in figure 3 c-d that were background subtracted using a momentum independent background but this was not clearly stated in the text. We apologize for this oversight and have amended figure 3 to only include raw data with background subtraction : here we show there is no significant difference between the two sets.

Reviewers' Comments:

Reviewer #1:

Remarks to the Author:

I would like to thank the Authors for satisfactorily addressing all the points raised in the previous reports. In my opinion the paper can now be published as a regular article in Nature Communications (without any further changes).

Reviewer #2:

Remarks to the Author:

Based on the measured Fermi surface volume, the authors determine the doping concentration to be $7 \pm 2\%$. Within one standard deviation, the system can be anywhere between 5% and 9%, which for electron-doped case ranges from the pseudogap/d-wave gap phase to the Fermi-liquid-like phase. Because of this uncertainty in the doping concentration, the only conclusion that can be drawn from the current study is: when doped enough in a clean way, the system enters a Fermi-liquid-like state also on the holed-doped side. I agree that this is nontrivial to show experimentally and it is done beautifully for the first time. However, the result is far from unexpected and unless I am missing something bigger, I tend to think this is not sufficient for publication in Nature communications.

Reviewers' Comments:

Reviewer #3:

Remarks to the Author:

I have little to add to the rather thorough review provided by Reviewer #1. I would like to suggest that the authors address the question of the doping level comparison they make with the Rh substituted systems. In particular, they argue that their 'intrinsically' hole doped samples at 7% K should be comparable to Rh doping at 4% because 'Cao et al have shown that each Rh introduces two holes in the IrO₂ planes' (quote from authors' response to first report of Reviewer #2) and have added text to this effect in the paper. I find this to be both misleading and perhaps incorrect. First, I am assuming that the Cao reference is Nat. Comm. 7:11367 (2016), in which the authors of that paper argue that Rh goes into the plane at light doping as Rh³⁺ rather than Rh⁴⁺, in effect generating an Ir⁵⁺ hole. I think this is reasonably well accepted in the community (and certainly would not be surprising to inorganic chemists). The Cao paper is silent on a 2:1 effective doping, and indeed my interpretation of that paper is that each Rh acts as a hole donor and accepts a single electron from a neighboring Ir. So, I suppose Nelson et al. may be invoking the effect of removing states from the upper Hubbard band upon hole doping into a 1/2-filled, gapped Mott insulator (e.g., in which the FS volume becomes twice the nominal hole concentration due to spectral weight transfer into the lower Hubbard band). If this is indeed what the authors mean, and I suspect it is, then they should more clearly articulate it. It would also if they could comment on Fig. 4 of Louat et al. PRB 97, 161109 (2018)) in the context of their own work. Here, the hole concentration is being measured from the Luttinger volume and shows effectively a 1:1 relationship between %Rh and hole concentration. If, on the other hand, they are arguing that Rh going into the plane generates 2 holes (rather than 2 states), then this needs to be squared with the literature (e.g., Fig. 4 of Chikara et al. PRB 95, 060407 (2017) that shows a 1:1 trend based on XAS rather than Luttinger volume estimates). Ultimately, I think it would be of value to the reader to have clarity from the authors on this point since they make comparison to Rh doping a key element of their manuscript.

In the end, a hole going into the plane due to Rh substitution in that plane is certainly going to be different than one deposited remotely by K on the Sr site. The authors have made an important step here in the iridate field, one that will be of importance not only to those working in that field but more generally to correlated electron oxide physics.

I recommend publication in Nature Communications with attention to the matter raised above.

We appreciate Reviewer 3's assessment that our work is significant to both the field of iridates and the broader study of correlated electron physics in oxides. We also would like to thank him/her for their careful comments regarding the hole doping of Sr_2IrO_4 when doped with Rh vs. K. Upon careful review of the literature, we agree with Reviewer 3's assertion regarding the number of doped holes introduced per Rh atom : *"I would like to suggest that the authors address the question of the doping level comparison they make with the Rh substituted systems. In particular, they argue that their 'intrinsically' hole doped samples at 7% K should be comparable to Rh doping at 4%..... if, on the other hand, they are arguing that Rh going into the plane generates 2 holes (rather than 2 states), then this needs to be squared with the literature..."*.

As a result, we have changed our text (lines 187-202) in the manuscript. In particular, we have changed our comparison of our 7% hole-doped $\text{Sr}_{1.93}\text{K}_{0.07}\text{IrO}_4$ sample to be with a 7% Rh-doped sample ($\text{Sr}_2\text{Ir}_{0.93}\text{Rh}_{0.07}\text{O}_4$) from the same reference (Louat *PRB* **97**, 161109), replacing the 4% Rh-doped sample from our earlier version. Fortunately, this change does not qualitatively affect our conclusions; there is still a 0.1 eV difference between the top of the $J_{\text{eff}} = 1/2$ band at $(\pi,0)$ for the Rh vs. K doped samples (as opposed to 0.25 eV from the previous version of the manuscript, when compared to the 4% doped sample). We have also removed the sentence : *"suggesting that the efficiency of each doped hole in $\text{Sr}_2\text{Rh}_{1-x}\text{Ir}_x\text{O}_4$ is lower than in our model"* from the manuscript.